# Serial, parallel and hierarchical decision making in primates

**Ariel Zylberberg[1,2]\*, Jeannette AM Lorteije[1,3]\*, Brian G Ouellette[1], Chris I De Zeeuw[4,5], Mariano Sigman[6], Pieter Roelfsema[1,7,8]\***

[1]Department of Vision and Cognition, Netherlands Institute for Neuroscience, Amsterdam-Zuidoost, Netherlands; [2]Howard Hughes Medical Institute, Columbia University, New York, United States; [3]Cognitive and Systems Neuroscience Group, Center for Neuroscience, Faculty of Science, Swammerdam Institute for Life Sciences, University of Amsterdam, Amsterdam, Netherlands; [4]Cerebellar Coordination and Cognition group, Netherlands Institute for Neuroscience, Amsterdam-Zuidoost, Netherlands; [5]Department of Neuroscience, Erasmus MC, Rotterdam, Netherlands; [6]Universidad Torcuato Di Tella, Buenos Aires, Argentina; [7]Department of Integrative Neurophysiology, VU University Amsterdam, Amsterdam, Netherlands; [8]Psychiatry Department, Academic Medical Center, Amsterdam, The Netherlands

**Abstract** The study of decision-making has mainly focused on isolated decisions where choices are associated with motor actions. However, problem-solving often involves considering a hierarchy of sub-decisions. In a recent study (Lorteije et al. 2015), we reported behavioral and neuronal evidence for hierarchical decision making in a task with a small decision tree. We observed a first phase of parallel evidence integration for multiple sub-decisions, followed by a phase in which the overall strategy formed. It has been suggested that a 'flat' competition between the ultimate motor actions might also explain these results. A reanalysis of the data does not support the critical predictions of flat models. We also examined the time-course of decision making in other, related tasks and report conditions where evidence integration for successive decisions is decoupled, which excludes flat models. We conclude that the flexibility of decision-making implies that the strategies are genuinely hierarchical.

\*For correspondence: ariel. zylberberg@gmail.com (AZ); j.a.m. lorteije@uva.nl (JAML); p.roelfsema@nin.knaw.nl (PR)

**Competing interests:** The authors declare that no competing interests exist.

## Introduction

The main aim of *Lorteije et al. (2015)* was to examine the neuronal correlates of hierarchical decision-making in the visual cortex. We trained monkeys to navigate through a decision tree with stochastic sensory evidence at branching points at two hierarchical levels, L1 and L2 (Figure 2 of *Lorteije et al., [2015]*). We measured the psychophysical kernels (*Ahumada, 1996*), which reflect the contribution of sensory information at different time-points to the decisions and recorded neuronal activity in visual cortical areas V1 and V4.

There are various strategies for solving this task. A decision maker could adopt a serial strategy, mentally tracing the curve from the fixation point and making a decision at each bifurcation until reaching one of the targets. Alternatively, evidence for the decisions could be integrated in parallel. In that case, the task might be represented as a 'flat' competition between four alternatives, or multiple decisions might first be considered independently before they are integrated to form a strategy. These models have been explored by *Lorteije et al. (2015)* and were reanalyzed in a companion paper by Hyafil and Moreno-Bote (H and M) (*Hyafil and Moreno-Bote, 2017*). The data

**eLife digest** Should you go for coffee with Jules, or go to the movie theater with Jim? Both options require you to make additional decisions, for example, which café would you go to, or what movie could you see? Many of the decisions we make in our daily lives feature a number of sub-decisions. Many of our day-to-day decisions have multiple layers of sub-decisions embedded within them that are not necessarily independent. Our opinions of the cafés in town and the movies showing at the theater may influence our decision over whom to spend the afternoon with.

In 2015, researchers at the Netherlands Institute for Neuroscience performed experiments in macaques to try to work out how the brain makes these decisions. The monkeys learned to choose between two visual stimuli (decision 1). The outcome of decision 1 determined whether the animals then had to make decision 2 or decision 3. The results suggested that the monkeys initially made all three comparisons independently and in parallel, before combining the evidence to select their overall strategy. This process is referred to as hierarchical decision-making. In the original analogy, one would compare the relative merits of Jules versus Jim, café A versus café B, and a horror movie versus a comedy at the same time before deciding what to do.

Other researchers have now reanalyzed the data from the original work using new computer simulations. This second analysis suggests that the results are more consistent with an alternative model of decision-making called a flat model, in which the brain compares all of the final options simultaneously (Jules + café A; Jules + café B; Jim + horror movie; Jim + comedy) before making a decision.

In response to these findings, Zylberberg et al. – who conducted the work reported in 2015 – reanalyzed the original data and re-ran the simulations. Zylberberg et al. argue that the flat model provides a poor fit to the original data. A new experiment with human volunteers suggests that modifying the task by adding even more decisions can lead to the different comparisons being made one after the other (in series) rather than all at the same time (in parallel), before the decision is made. This is difficult to explain with a flat model. Zylberberg et al. argue that these findings confirm the original conclusion that the monkeys use a hierarchical strategy. Moreover, the new results expose a previously unknown limit in the number of decisions that the brain can evaluate at any one time. If this limit is exceeded, decision-making becomes serial.

Future studies can build on these findings by further exploring the limits of parallel decision-making, which may help us to understand how the brain is able to make multiple decisions while keeping the future consequences in mind.

of *Lorteije et al. (2015)* ruled out serial strategies, which wrongly predict that the psychophysical kernel for the L2 (level 2) decision should be protracted relative to L1 and that evidence integration for L1 (level 1) precedes evidence integration for L2.

*Lorteije et al. (2015)* also presented evidence against 'flat' models with a single accumulator per motor action, which predict that L1 difficulty should affect L2 accuracy. When the L1 decision is easy, strong evidence for L1 causes the decision variable to be close to the bound so that less evidence is required to make the L2 decision. We confirmed this intuition by fitting a simple (3-parameters) instantiation of a 'flat' model to the behavioral data. As in most models of sensory decision making (*Smith and Ratcliff, 2004*; *Gold and Shadlen, 2007*)—but contrary to the model by H and M—we assumed that the evidence is integrated without self-excitation (or leak) until reaching a criterion or bound. The model fits revealed that the decision-termination bound was low, which led to short integration times. This observation was supported by a model-free analysis of the psychophysical kernels, showing that the monkey's choices were mainly influenced by the earliest luminance samples. Later samples made only a small contribution to the decision even though the monkeys had to wait at least 500 ms before they could report their decision. Unlike the monkeys, our instantiation of the flat model exhibited an influence of L1 difficulty on L2 accuracy, which led us to reject the flat model.

H and M challenge our conclusion about flat models (*Hyafil and Moreno-Bote, 2017*), putting forward an 8-parameter version of the leaky-competing accumulator model (LCA [*Usher and*

*McClelland, 2001*; *Tsetsos et al., 2012*]), which at first sight captures many aspects of the monkeys' behavior. We therefore consider this model an interesting addition to the literature on the neural mechanisms for hierarchical decision-making. In H and M's model, the task is represented as a flat competition between the four possible motor actions. Each alternative (i) integrates evidence simultaneously from L1 and L2, (ii) is endowed with self-excitation, establishing a positive feedback loop, and (iii) is inhibited by a global process that affects all races equally.

Here, we will first examine the manner in which H and M avoided the influence of L1 difficulty on L2 accuracy and conclude that it is neither compatible with the behavioral data nor with the neurophysiological data. We then examine more general properties of flat models and conclude that they lack the flexibility to account for decision making in hierarchical tasks.

H and M's model reduced the influence of L1 difficulty on L2 accuracy by implementing high bounds, strong recurrent excitation and sensory noise that does not scale with the strength of the evidence. Self-excitation was needed in H and M's model to explain why the monkeys assigned high weights to the early samples in our task. Without self-excitation, models with high bounds produce flat psychophysical kernels, because they assign equal weights to many samples before committing to a choice (*Brunton et al., 2013*).

Unfortunately, the self-excitation in the model of H and M causes unstable activity. This can be seen in H and M's Figure 3C where activity appears to follow an exponential time-course, which is incompatible with the findings of *Lorteije et al. (2015)* and other neurophysiological results (e. g. *Kiani et al., 2008*). One possibility to stabilize the firing rates in H and M's model would be to use an attractor network (*Wang, 2002*; *Wong and Wang, 2006*). However, attractors act as bounds and would increase the dependency of L2 accuracy on L1 difficulty. Furthermore, in primates and rodents evidence accumulation can occur without loss of information about the later samples (*Kira et al., 2015*; *Kiani et al., 2013*; *Brunton et al., 2013*), which is incompatible with strong self-excitation.

We simulated the two models proposed by H and M and observed that the high bounds combined with self-excitation did not suffice to prevent the interaction between L1 and L2. The first model had infinite bounds collapsing after 500 ms, and the second one had high constant bounds, which were not reached on a large fraction of trials (*Figure 1—figure supplement 1*). We investigated whether the accuracy of the L2 decision depended on the accuracy of the L1 choice. In both models, L2 decisions were less accurate if the L1 decision was correct than if it was erroneous. This decrease in accuracy is visible as a flattening of the psychometric functions (*Figure 1*). We reanalyzed the data of *Lorteije et al. (2015)* and observed much weaker interactions between the two

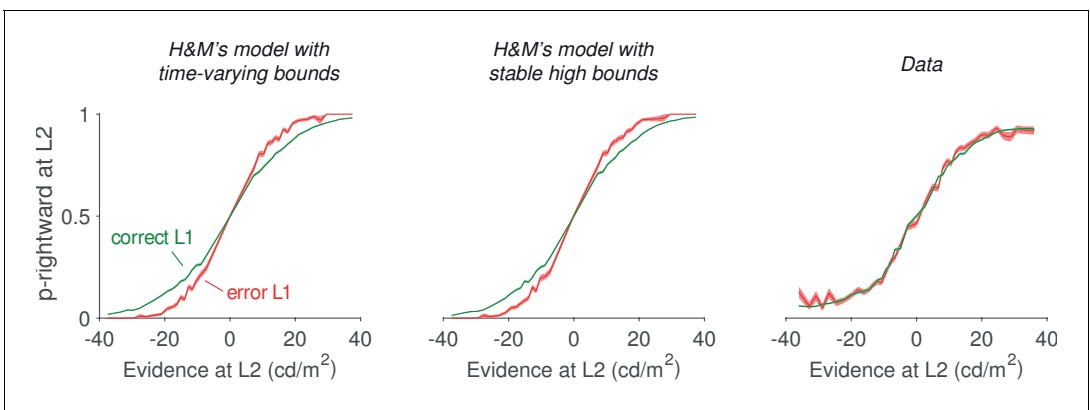

**Figure 1.** Interaction between the accuracies at L1 and L2. We computed the psychometric function for the L2 decision for trials with a correct (green traces) or erroneous (red) decision at L1. Left, H and M model with infinite bounds collapsing after 500 ms. Middle, H and M model with high bounds that are stable. Right, data of the monkeys in *Lorteije et al. (2015)*. Green/red regions, s.e.m.

The following figure supplement is available for figure 1:

**Figure supplement 1.** Reaction times for model and data.

decisions (*Figure 1*). Relaxing some of H and M's assumptions (e.g. including noise that scales with the strength of the evidence [*Zylberberg et al., 2016*]) caused even stronger crosstalk.

We conclude that the models of H and M fail to account for the behavioral data. We next reanalyzed our neurophysiological data to test H and M's model more directly. Specifically, the authors stated that the model predicts that 'selection signals at levels two are only influenced by information provided at level two branches [...] and not by information provided at level 1'. To test this prediction, we grouped the trials into four categories based on the strength of evidence at L1 and examined the activity elicited by L2 branches in V4 (*Figure 2A*).

Our V4 recordings revealed that trials with strong evidence at L1 boosted the representation of the TT branch, suppressed the representation of the DT and DD branches (linear regression, all Ps $< 10^{-3}$) (*Figure 2A*). These distinct effects of L1 evidence on the four L2-branches are incompatible with any model in which the interactions between decisions are governed by a single source of inhibition, including the one by H and M. Indeed, when we conducted the same analysis in H and M's model, we observed that stronger L1 evidence decreased the activity for the DT and DD branches and increased the activity for both the TT and TD branches (*Figure 2B*). A possible explanation for the difference in activity elicited by the TD branch, which is suppressed in H and M's model but not in the data, is that it was already suppressed by the selection of the TT branch. These more complex and local interactions between the L1 and L2 decisions are consistent with a genuine hierarchical decision making process and are incompatible with flat models with a single source of inhibition.

Hyafil and Moreno-Bote criticized *Lorteije et al. (2015)* by stating that ad-hoc assumptions need to be made to explain the interactions between local selection signals in a hierarchical model. Indeed, hierarchical models imply that there are mechanisms to enable interactions between the different decisions. For example, *Lorteije et al. (2015)* demonstrated that the relative difficulty of the L2 decisions biases the L1 choice, an effect that is not only visible in the animals' behavior but also in the modulation of firing rates in visual cortex. Because H and M's global inhibitory process is refuted by the data, it follows that another mechanism should play an equivalent role. It is conceivable, for example, that there is a process that compares the confidence in the two L2 decisions and then biases the L1 decision. *Lorteije et al. (2015)* implemented a simple version of this mechanism where the confidence in the two L2-decisions was evaluated to select the target that maximizes reward. Such a role of confidence is supported by previous studies demonstrating that the confidence in one decision can guide another one (*Kepecs et al., 2008*; *van den Berg et al., 2016*; *Kiani and Shadlen, 2009*). We believe that elucidating the interactions between local selection signals is an important topic for future research. Importantly, these interactions between local selection signals do not provide evidence in favor of or against flat models. They are equally compatible with hierarchical models.

H and M also claim that their two models are more parsimonious than the models in *Lorteije et al. (2015)*. We disagree with this parsimony claim. The models proposed by H and M contain a large number of parameters and have the same shortcomings as the simpler flat model that was explored in *Lorteije et al. (2015)*. We are also concerned that the results by H and M critically depend on (1) infinite bounds that collapse rapidly, which have not been observed in neurophysiology, (2) noise that does not scale with the strength of the evidence, and (3) strong self-excitation that causes a suboptimal decision making process and exponentially-increasing firing rates. More importantly, when testing the models by H and M, we found that they cannot explain the lack of an influence of L2 accuracy on L1 sensitivity, nor the absence of an influence of L1 luminance strength on the TD branch, and as such they fail to account for the behavioral and neurophysiological results in the hierarchical decision making task of *Lorteije et al. (2015)*.

We will now turn to a more general problem that flat models incur when they are applied to hierarchical decision-making tasks. Flat models make all local decisions at the same time, because there is a single bound that governs them all. In contrast, previous curve-tracing studies presented evidence for serial decision making and we will here illustrate that shifts between parallel and serial strategies can occur. Indeed, under many conditions, curve-tracing invokes a serial operation, which is implemented in the visual cortex by the propagation of enhanced neuronal activity along a target curve (*Pooresmaeili and Roelfsema, 2014*). This propagation of enhanced neuronal activity has a correlate in psychology: observers gradually spread object-based attention over the relevant curve (*Houtkamp et al., 2003*). In one previous study (*Pooresmaeili and Roelfsema, 2014*) monkeys had to mentally trace a target curve that was connected to a fixation point (*Figure 3*). The target curve

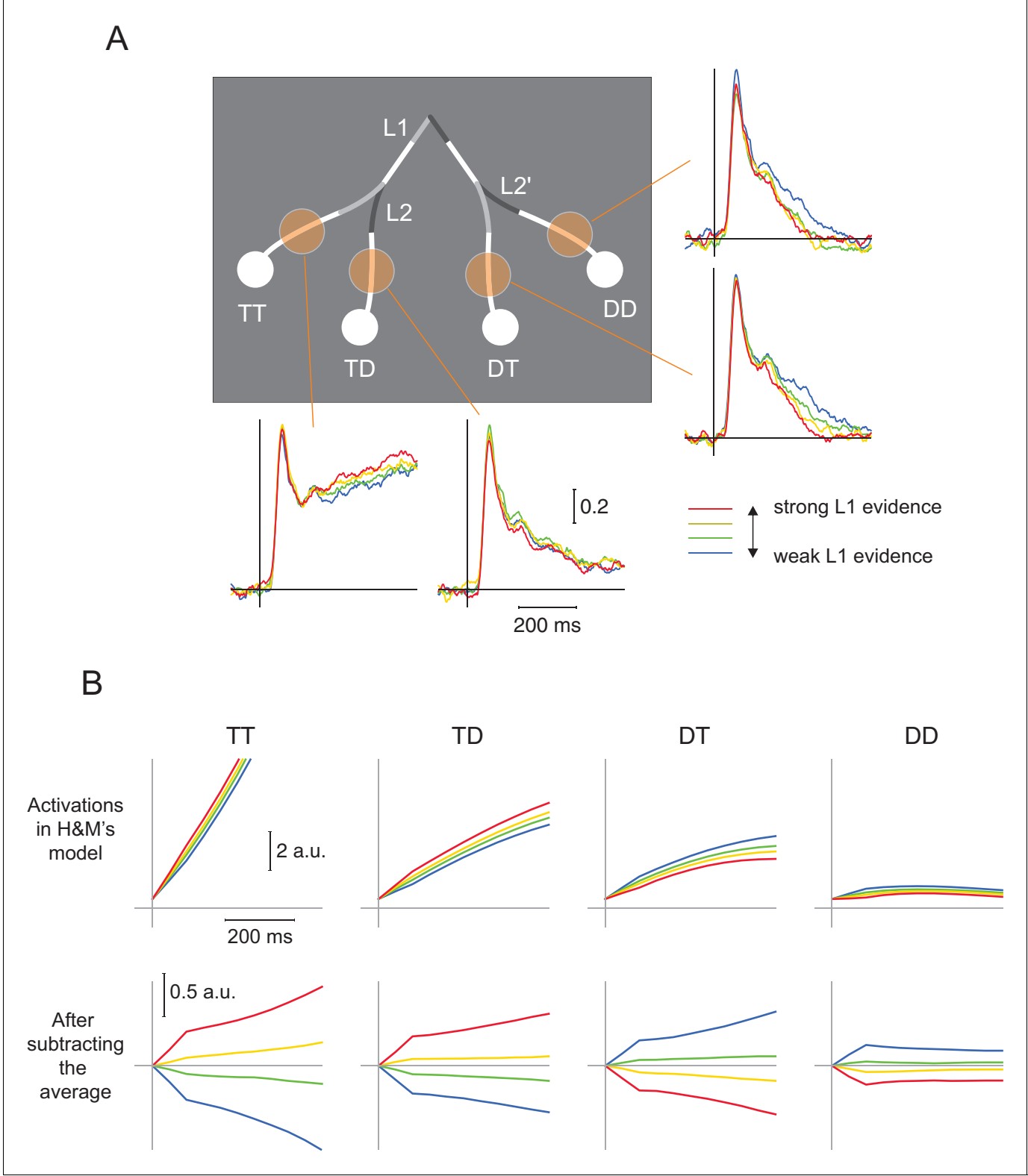

**Figure 2.** Influence of strength of evidence at L1 on V4 activity elicited by the L2 and L2' branches in V4 and in the model of H and M. (**A**) In the data, high L1 signal strength increases the representation of TT, suppresses the representations of DT and DD, but has little impact on TD. This finding is incompatible with flat models that include only one source of inhibition. (**B**) Upper panel, Same analysis conducted for the activity in the model of H and M. Note the steep rise in activity, which is very different from the neuronal activity in area V4. High L1 signal strength increased activity elicited by

*Figure 2 continued on next page*

*Figure 2 continued*

both TT and TD, and suppressed DT and DD. Because of the rectification (firing rates cannot take negative values), L1 evidence has the weakest influence on the DD branch. Lower panels, same data after the average activity was subtracted. Both models by H and M yielded similar results.

either crossed with a distractor curve or the curves did not intersect. We found that neuronal activity elicited in V1 by L1-segments of the target curve near the fixation point was enhanced before the activity elicited by the L2-segments beyond the location of the possible crossing. Thus, the enhanced V1 activity first reflects the L1-decision about the connection at the fixation point and only later reflects the L2-decision about the crossing. This result refutes flat models that implement a race between the four possible interpretations of the stimulus (i.e. one accumulator for each of the four configurations on the left of *Figure 3*).

Why is hierarchical decision-making parallel in some situations (*Lorteije et al., 2015*) and serial in others (*Figure 3*)? Multiple factors may be at play. The first factor is the configuration of the stimulus. The crossings in the stimulus of *Figure 3* do not provide independent evidence about the target or distractor status of the L2 branches (the problem has an 'exclusive-or' structure), whereas in the task of *Lorteije et al. (2015)* local evidence at L2 permitted discarding branches as distractors before the L1 decision was taken.

A second factor that may influence the degree of seriality is the number of decisions that have to be taken at the same time. We examined this possibility in a new experiment, which tested the capacity of parallel decision-making in human observers. We inferred the time-course of evidence accumulation by adding stochastic luminance evidence to branching points so that we could estimate the subjects' psychophysical kernels, similarly to *Lorteije et al. (2015)*. *Figure 4a* illustrates our findings with a hierarchical decision-making task with two levels. At the L1-decision, subjects had to determine which branch connected to the fixation point (left or right) had the brightest image elements and they made a second decision (highest luminance up or down) at the relevant L2 branching point. Evidence for these decisions accumulated largely in parallel, just as in *Lorteije et al. (2015)*. However, the decision-making process switched to a more serial mode when we added a third level (L3) so that the total number of decisions increased to seven (*Figure 4b*). Now most of the evidence for L1 accumulated before the evidence for L2, which, in turn accumulated mostly before the evidence for L3. The implication is that there is a limited capacity for making decisions in this task. Subjects can accumulate evidence for between 3 and 7 decisions in parallel, although the exact capacity for parallel decision making remains to be determined. These results also imply that there are conditions under which evidence for L1 and L2 can accumulate independently. It is therefore parsimonious

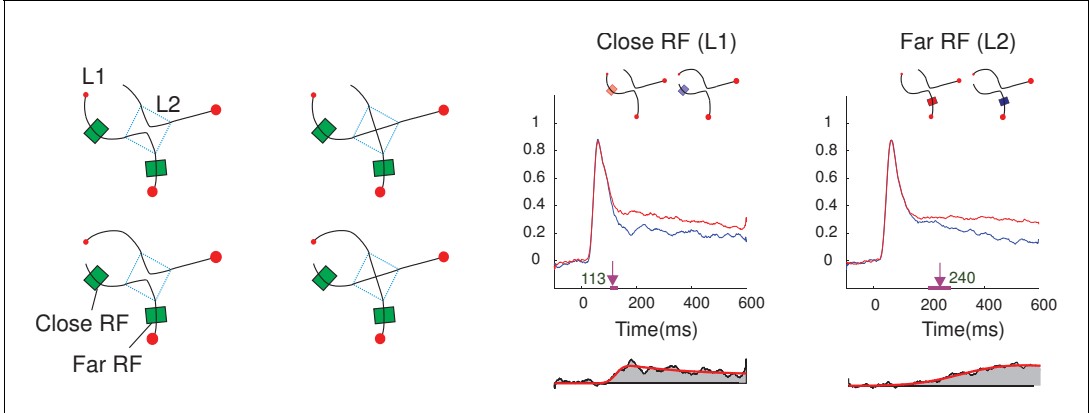

**Figure 3.** Curve-tracing task with possible intersections. V1 activity elicited by a target curve was stronger than that elicited by a distractor curve. The latency of this enhanced activity reveals a serial decision process, as the selection signal of neurons with a receptive field near the fixation point (close RF, L1-decision) preceded the selection signal of neurons with a receptive field beyond the crossing (far RF, L2-decision). The lower panels on the right represent the selection signals (target minus distractor response). Reprinted with permission from *Pooresmaeili and Roelfsema (2014)*[18].

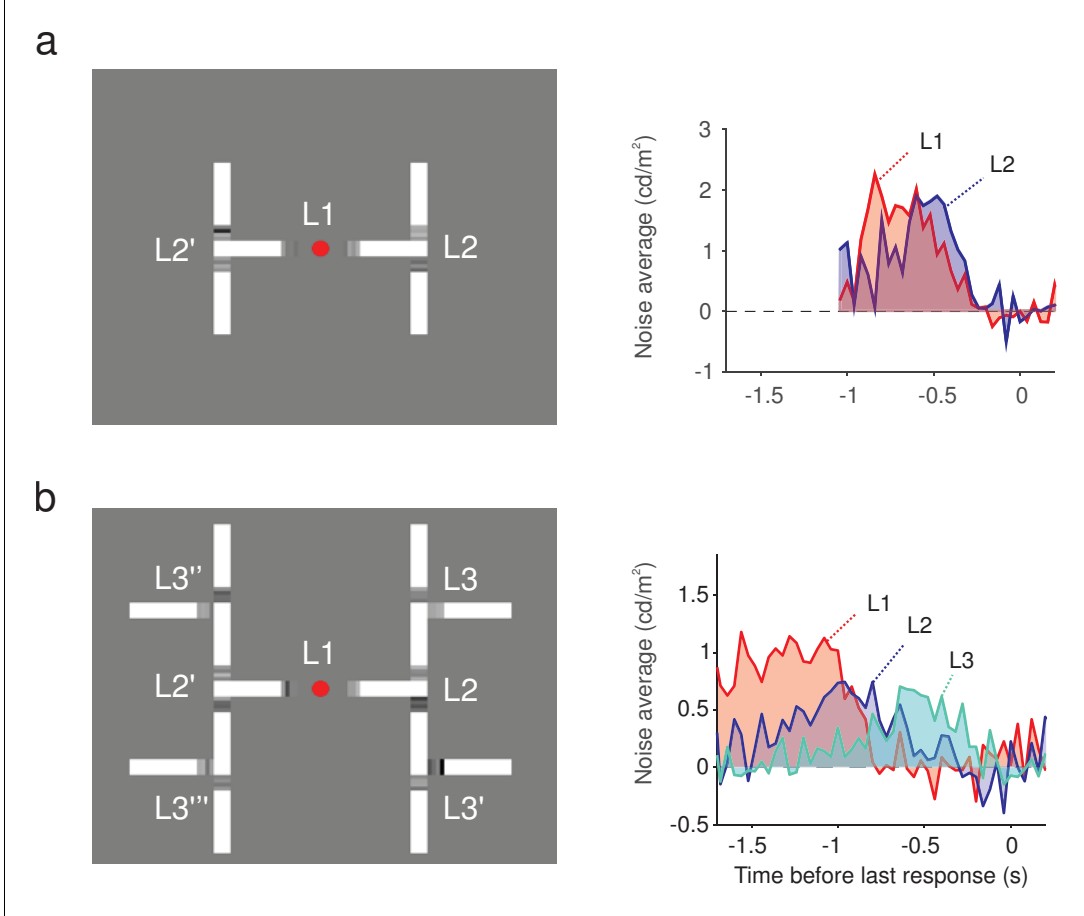

**Figure 4.** Integrating evidence in a hierarchical decision making task with two (a) and three levels (b). (a) Left, at every bifurcation, subjects had to choose the branch with the higher luminance. At L1 they chose left/right and at L2 they chose up/down. Right, the psychophysical kernel for L1 (red) and L2 (blue), which represents the influence of luminance fluctuations at different time points aligned to the second response. (b) Adding a L3 decision caused a more serial decision process and decreased the overlap between the kernels for the L1 and L2 decisions. Panel b was reproduced with permission from *Zylberberg et al. (2012)*.

to assume that these separate accumulators also contribute to the task with only two decisions, in accordance with a genuine hierarchical decision making process. Thus, flat models can also not account for results in tasks that are intimately related to the task studied by *Lorteije et al. (2015)*. Furthermore, flat models need to reserve a single accumulator for every configuration of successive decisions (e.g. eight accumulators for the task of *Figure 4b*), causing them to scale badly if decision trees become elaborate.

A third factor that can influence the seriality of decision making is practice. Prolonged training on a fixed set of problems is likely to result in more efficient decision making process, where multiple sources of evidence are efficiently considered and combined to form a strategy. We note, however, that the subjects of the tasks of *Figures 3* and *4b* performed many thousands of trials but that their decision-making process remained serial.

These results, taken together, indicate that hierarchical decision-making is a highly flexible process during which evidence can be integrated by separate accumulators, allowing subjects to adapt their strategy to the task at hand. Parallel accumulation can only occur if it is permitted by the structure of the task (i.e. if it does not have the exclusive-or structure) and if the subject's capacity for parallel decision making is not exceeded. Flat models lack the required flexibility.

We conclude that critical predictions of the models of Hyafil and Moreno-Bote are not supported by the data and, more generally, that flat models fail to account for any situation where decision making is serial (*Figure 3*). Our finding in humans that adding L3 branches increases seriality at L1

and L2 (*Figure 4*) implies that the machinery to take independent decisions at L1 and L2 is always in place. To further elucidate the neuronal mechanisms for hierarchical decision making, future studies could record from neurons with persistent activity, in areas like LIP or FEF, using variants of the task studied by Lorteije et al. They could also focus on the neuronal mechanisms that improve the interactions between successive decisions when subjects become proficient in a task.

## Materials and methods

### Monkey electrophysiology

All experimental procedures complied with the National Institutes of Health Guide for Care and Use of Laboratory Animals, and were approved by the Institutional Animal Care and Use Committee of the Royal Netherlands Academy of Arts and Sciences. Detailed methods can be found in the original publication (*Lorteije et al., 2015*). Briefly, multi-unit activity was collected from multi-channel electrode arrays implanted in the visual cortex (area V1 and V4) of 2 monkeys, while these animals performed a hierarchical curve tracing task. For the analysis in *Figure 2A*, we grouped trials in quartiles of L1-evidence strength. For each trial, we averaged the luminance of the target and distractor segments at L1 over the first two samples, and then computed the difference. We then classified trials into quartiles, including only correct trials, independently for each of the three difficulty levels. *Figure 2* shows the average firing rate (normalized as in *Lorteije et al., 2015*) within each quartile for receptive fields located at different branches. The averages are from 31 sites, and they were obtained in a total of 19 recording sessions.

### Human decision-making in tasks with two- and-three level hierarchies

The three-level task displayed in *Figure 4* was reproduced from *Zylberberg et al. (2012)*. The data from the two-level hierarchical task has not been published before.

Three subjects had to trace a curve that bifurcated twice, completing between 1825 and 2534 trials each. Subjects had to make a left/right decision at L1, and an up/down decision at L2, reporting their decisions with different key-presses. As in *Zylberberg et al. (2012)*, luminance samples were perturbed with additive Gaussian noise ($\sigma$ = 10 cd/m$^2$), the background luminance was 50 cd/m$^2$ and the mean luminance of the target was adjusted online with a Quest procedure to keep performance at ~75% correct. The psychophysical reverse correlation analysis was conducted as explained in the Supplemental Information of *Zylberberg et al. (2012)*.

## Acknowledgements

PRR was supported by NWO (ALW grant 823-02-010) and the European Union (the Human Brain Project Grant Agreement No. 720270 and ERC Grant Agreements n. 339490 'Cortic_al_gorithms' and n. CCC).

## Additional information

### Funding

| Funder | Grant reference number | Author |
| --- | --- | --- |
| Nederlandse Organisatie voor Wetenschappelijk Onderzoek | 823-02-010 | Pieter Roelfsema |

The funders had no role in study design, data collection and interpretation, or the decision to submit the work for publication.

### Author contributions

AZ, Conceptualization, Data curation, Software, Investigation, Visualization, Writing—original draft, Writing—review and editing; JAML, Conceptualization, Data curation, Investigation, Writing—original draft, Writing—review and editing; BGO, Conceptualization, Data curation, Software, Investigation; CIDZ, Conceptualization, Supervision, Funding acquisition, Writing—review and editing; MS, Conceptualization, Resources, Supervision, Funding acquisition, Investigation, Writing—review and

editing; PR, Conceptualization, Resources, Supervision, Funding acquisition, Investigation, Writing—original draft, Project administration, Writing—review and editing

**Author ORCIDs**
Ariel Zylberberg, http://orcid.org/0000-0002-2572-4748
Pieter Roelfsema, http://orcid.org/0000-0002-1625-0034

**Ethics**
Human subjects: We obtained ethical approval from the ethics committee at the University of Amsterdam and informed consent in writing from the subjects before the experiments.
Animal experimentation: All experimental procedures complied with the National Institutes of Health Guide for Care and Use of Laboratory Animals, and were approved by the Institutional Animal Care and Use Committee of the Royal Netherlands Academy of Arts and Sciences under protocol number NIN11.19 "attentional and reward modulation during multiple perceptual decision making".

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
