## [Decision Letter]

Thank you for submitting your article "Serial, parallel and hierarchical decision making in the visual cortex" for consideration by *eLife*. Your article has been reviewed by a Reviewing Editor and Timothy Behrens as the Senior Editor.

The reviewers have discussed the reviews with one another and the Reviewing Editor has drafted this decision to help you prepare a revised submission.

Summary:

These two papers present an enlightened and useful discussion about the interpretation of results previously published by Lorteije and colleagues. In that prior study, monkeys performed a task that required them to make a saccadic eye movement to the appropriate endpoint of a visual branching pattern with three bifurcations: one at the top, then two others under each branch, resulting in four distinct endpoints. Each bifurcation had a modulating luminance cue at each branch that determined the correct path: always choose the brighter (on average) branch. Through analyses of behavior and the activity of neurons in cortical areas V1 and V4 whose receptive fields corresponded to the locations of the luminance cues, a primary conclusion from that study was that the monkeys solved the task in a hierarchical manner, with decisions about the top- and lower-level branches first occurring in parallel, then combined for a final choice.

That landmark paper spawned several interesting discussions in the field. Hyafil and Moreno-Bote's technical comment encapsulates one of the critical lines of discussion about whether it is possible to truly distinguish a hierarchical decision-making process from a flat process based on the experimental data of Lorteije and colleagues. It is a critical and complex question. The reply by Zylerberg and colleagues adds further to the discussion, presenting several counter-arguments to the claims of Hyafil and Moreno-Bote.

All three reviewers were impressed by the tone and content of the submissions and agree that both represent worthwhile contributions to the literature. As noted by one of the reviewers, this kind of debate is "very valuable and generally underappreciated."

The above comments are included, verbatim, in our decision letters to both groups of submitting authors. We also will now make both initial submissions available to both groups, so that any potential revisions can fully take into account the claims made in the other submission. We will then allow for one more iteration: if and when we receive revised submissions and deem them appropriate, we will then again make each available to the other group for further revisions and clarifications.

Below are summaries of the discussions among the reviewers that are specific to your submission.

Essential revisions:

The reviewers agreed this paper presents several important arguments and analyses that support the idea of a hierarchical model. However, they also raised several concerns:

1) The paper may be stronger and more accessible if it was reorganized such that the rebuttals of the arguments for the flat model that were made by Hyafil and Moreno-Bote were addressed first. That is, more directly address whether the data in the original submission by Lorteije et al. exclude a flat model with inhibition.

2) It also would be useful treat more directly the claim by Hyafil and Moreno-Bote that the hierarchical model requires an "extra modulatory signal" from L2 to L1 "that must be carefully tuned." This brings up the more general point that, as the authors are well aware, sensory neural responses provide only indirect evidence for the mechanism of the decision-making process. In the absence of a clear understanding about the nature of choice-related feedback to V1 and V4 one should interpret the neural responses with caution. Making this point more explicitly would be useful.

3) Although the other data presented here are interesting, there were questions about their relevance to the main argument. For example, the authors should be more explicit about how the gradual spread of object-based attention in the "mental curve tracing" tasks relate to hierarchical decision-making. The should also clarify exactly why that result "cannot be explained by 'flat' models that implement a race between the four possible interpretations of the stimulus." Likewise, the data from the three-branch task is interesting, and under those conditions subjects likely rely on a serial and hierarchical decision-making process based on accumulation of evidence that is largely (albeit not necessarily entirely, as the authors seem to imply) independent at different levels of the decision tree. However, does this result necessarily imply that a decision tree with two branching points is complex enough to necessitate a hierarchical decision? It seems quite possible that subjects' strategy shifts from a flat process to a hierarchical process as the number of branching points increases.

4) The assertion that a flat model predicts either unrealistically long reaction times or a dependence of L2 decisions on L1 stimulus strength raised several concerns. First, the task did not have a reaction time design -- the monkey had to view the stimulus for 500ms before responding. It is unclear whether subjects curtailed their decisions after the fixation point turned off, continued to accumulate evidence toward a decision bound, or used a mixture of those strategies. The very weak dependence of RT on stimulus strength (only ~100ms RT difference between the weakest and strongest stimuli) make them particularly difficult to interpret in terms of decision models. Second, additional mechanisms, such as a collapsing decision bound (or urgency) might make a flat model compatible with the RT data. Is a quantitative match between the model and data truly impossible?

[Editors' note: further revisions were requested prior to acceptance, as described below.]

Thank you for submitting your article "Serial, parallel and hierarchical decision-making in the visual cortex" for consideration by *eLife*. Your article has been evaluated by a Reviewing Editor and Timothy Behrens as the Senior Editor.

We received revised manuscripts from both your group and from Hyafil and Moreno Bote. We asked them to respond to a few remaining issues, which they have done. As we indicated to them, we are now forwarding to you the newest version of their manuscript, to give you an opportunity to revise and respond accordingly. Once we hear back from you, we then will share both papers with both groups, but at that point, if any further changes are desired, they have to be essential and very well justified.

There are two points in particular that we think might benefit from further discussion/analyses in your paper. Specifically:

1) As you have indicated, the behavioral data show that stimulus difficulty at the L1 branch did not influence performance at the L2 branch. In the original Lorteije et al. paper, a flat model without lateral interactions did not reproduce this result. Hyafil and Moreno Bote then showed that it could be reproduced with a flat model with lateral interactions and rectification. You have pointed out that implementation required very high decision bounds that would have produced unrealistic RTs. Hyafil and Moreno Bote now produce a version of the model that reproduces the L1 dependence and realistic mean RTs, using a kind of "collapsing bound" that is infinite (unreachable) for the first several samples that appears to reasonably account for the task design that allowed responses only after 500 ms of stimulus viewing.

2) Both papers discuss the new analysis by Zylberberg et al. showing that the strength of evidence at L1 affects V4 activity elicited by the L2 and L2' branches. Both papers are in agreement with the finding but disagree with the interpretation. Part of the disagreement appears to stem from misunderstandings from some comments in an earlier version of the Hyafil and Moreno Bote manuscript. Specifically, as quoted by Zylberberg et al., Hyafil and Moreno Bote state in Appendix A that the flat model predicts that "selection signals at levels 2 are only influenced by information provided at level 2 branches […] and not by information provided at level 1." The new analyses by Zylberberg et al. clearly contradict this prediction. However, in their main text, Hyafil and Moreno Bote stated that "this observation is most compatible with the flat model and by itself rules out the hierarchical model that relies on complete neural segregation of integration of L1 and L2 evidence." We asked Hyafil and Moreno Bote to reconcile these statements. In the new manuscript, they provide the following comment: "At this point, an important distinction has to be made in the hierarchical model between localized activations, which indeed mix evidence from both branches, and selection signals at L2, extracted by looking at the difference between two units in the same L1 branch, and which as shown above are largely insensitive to L1 signals."

---

## [Author Response]

*Essential revisions:*

*The reviewers agreed this paper presents several important arguments and analyses that support the idea of a hierarchical model. However, they also raised several concerns:*

*1) The paper may be stronger and more accessible if it was reorganized such that the rebuttals of the arguments for the flat model that were made by Hyafil and Moreno-Bote were addressed first. That is, more directly address whether the data in the original submission by Lorteije et al. exclude a flat model with inhibition.*

We thank the reviewers for this suggestion and we have reorganized our manuscript accordingly.

*2) It also would be useful treat more directly the claim by Hyafil and Moreno-Bote that the hierarchical model requires an "extra modulatory signal" from L2 to L1 "that must be carefully tuned." This brings up the more general point that, as the authors are well aware, sensory neural responses provide only indirect evidence for the mechanism of the decision-making process. In the absence of a clear understanding about the nature of choice-related feedback to V1 and V4 one should interpret the neural responses with caution. Making this point more explicitly would be useful.*

We now discuss this point in the revised manuscript. Our results indeed demonstrate that the difficulty of the L2 decisions (or decision confidence) biases the L1 choice. However, this experimental finding is not incompatible with hierarchical decision making strategies. Previous studies have shown that monkeys (and rats) can use confidence in one decision to guide other decisions, for instance how long to wait for a reward (e.g., Kepecs et al., 2008) or whether to opt out of a difficult choice (e.g., Kiani and Shadlen, 2009). The implication is that there must be mechanisms that carry information from one decision to a later one. In these cases, interactions occurred between decisions separated in time, which is beyond the scope of “flat” accumulator models.

We believe that it would be premature to commit to a specific mechanism that mediates the interactions between local decisions, because existing data are compatible with several possibilities. The model of H and M proposes one such mechanism, where the interaction between sub-decisions is mediated by global inhibition during a ‘flat’ competition between the alternative motor actions. As we argue in our response, this mechanism cannot explain situations where decisions become serial, and it therefore lacks the required flexibility to account for the interactions between sub-decisions that are observed more generally. Furthermore, the global inhibitory process proposed by H and M is also incompatible with the re-analysis of the neural data of Lorteije et al. as presented in Figure 2 of our response.

We believe that the different possible mechanisms that might enable interactions between local decisions deserve to be tested in future modeling studies and neurophysiological work. We modeled one such alternative mechanism that can account for the influence of the relative difficulty of the two L2 decisions on the L1 decision. In a diffusion model, the state of accumulated evidence (the decision variable, D) at a particular time t is monotonically related to confidence (e.g. Kiani and Shadlen 2009). We therefore used the difference between the absolute values of the DVs of the two L2 decisions (ΔDt) as an index of the relative difficulty of the L2 decisions. We then implemented an influence of ΔDt on the L1 decision in the hierarchical model of Lorteije et al., 2015 by adding an extra term to the evidence at L1. This term equaled (ΔDt−ΔDt−1),where k is a scaling constant and t and t−1 are successive time steps. As illustrated in Figure 5, the relative difficulties of the two L2 decisions influenced the L1 decision just as in the behavior of the monkeys. Thus, the influence of L2 difficulty on L1 choice behavior can be incorporated into our hierarchical model as an additive term (which did not require “careful tuning” other than adjusting one parameter, *k*). We emphasize, however, that given the large number of alternative mechanisms that could potentially mediate the interactions between local decisions, we prefer to test them in future work rather than prematurely committing to one of them.

Our revision states that the global inhibition proposed by Hyafil and Moreno-Bote is only one of a number of possible mechanisms to mediate the interactions between local decisions and we mention our preference to stay uncommitted.

Author response image 1.The decision at L1 is biased by the relative difficulty of the two L2 decisions.Similar to the results of Lorteije (2015), L2 affects L1 only when the first decision is difficult.**DOI:**
http://dx.doi.org/10.7554/eLife.17331.008

The reviewers also mention that care should be taken when interpreting the role of neuronal signals in V1 and V4 for decision making. We fully agree with this point. In our original publication (Lorteije et al., 2015, page 6 paragraph 1) we reflected on the possible role of the modulation of neuronal firing rates in visual cortex and indicated that we do not have prove for a causal role of these signals in the decision making process. This is why we chose to use the neutral term ‘selection signal’ so that we could remain uncommitted about the role of these signals. Given the limited number of words of our rebuttal and because this issue was clearly addressed in our original paper, we did not repeat the discussion of the putative role of these selection signals in the decision process in our response to Hyafil and Moreno-Bote. These authors did not have concerns about the validity of measuring selection signals in visual cortex but they rather called our hierarchical decision-making model into question.

*3) Although the other data presented here are interesting, there were questions about their relevance to the main argument. For example, the authors should be more explicit about how the gradual spread of object-based attention in the "mental curve tracing" tasks relate to hierarchical decision-making. The should also clarify exactly why that result "cannot be explained by 'flat' models that implement a race between the four possible interpretations of the stimulus." Likewise, the data from the three-branch task is interesting, and under those conditions subjects likely rely on a serial and hierarchical decision-making process based on accumulation of evidence that is largely (albeit not necessarily entirely, as the authors seem to imply) independent at different levels of the decision tree. However, does this result necessarily imply that a decision tree with two branching points is complex enough to necessitate a hierarchical decision? It seems quite possible that subjects' strategy shifts from a flat process to a hierarchical process as the number of branching points increases.*

We thank the reviewers for pointing out that we had made it insufficiently clear why cases of more serial decision making are relevant for the issues raised by Hyafil and Moreno-Bote. We now clarify that more serial decision making strategies reveal a general insufficiency of the type of flat models proposed by Hyafil and Moreno-Bote. In a flat model, there is only one accumulator for each motor response. These models cannot account for cases where evidence for the L1 decision accumulates before evidence for the L2 decisions. One example of such a more serial decision making strategy has been illustrated in Figure 3 of our rebuttal, where information about the L1 target branch connected to the fixation point accumulates before evidence about the L2 target branch behind the possible location of an intersection.

Furthermore, the data of Figure 4 demonstrate that subjects can flexibly shift between parallel and more serial hierarchical decision making strategies in stimulus configurations equivalent to those studied by Lorteije et al., 2015. The implication is that separate accumulation processes must exist for each of the individual local decisions. Hierarchical models can shift from parallel to more serial strategies by changing the timing of the individual decisions and the later integration process.

In contrast, flat models propose a single accumulator for each of the possible interpretations of the stimulus so that all the evidence must accumulate at the same time. This machinery proposed by Hyafil and Moreno-Bote would have to be replaced by a different neuronal substrate to permit the independent integration of evidence for local decisions once the L3 decision has been added. Hence such a flat model is not parsimonious. In contrast, hierarchical models flexibly switch between parallel and more serial strategies, strengthening the claim that hierarchical models also apply to situations with fewer local decisions.

*4) The assertion that a flat model predicts either unrealistically long reaction times or a dependence of L2 decisions on L1 stimulus strength raised several concerns. First, the task did not have a reaction time design -- the monkey had to view the stimulus for 500ms before responding. It is unclear whether subjects curtailed their decisions after the fixation point turned off, continued to accumulate evidence toward a decision bound, or used a mixture of those strategies. The very weak dependence of RT on stimulus strength (only ~100ms RT difference between the weakest and strongest stimuli) make them particularly difficult to interpret in terms of decision models. Second, additional mechanisms, such as a collapsing decision bound (or urgency) might make a flat model compatible with the RT data. Is a quantitative match between the model and data truly impossible?*

The central question here is whether the behavioral evidence does or does not support the view of the crossing of a decision bound. The main signature of evidence integration towards a relatively stable bound (that does not quickly collapse) is a variable reaction time that depends on the difficulty of the stimulus. This is precisely what we observed. Although we did impose a minimal viewing duration of 500 ms before the monkey was allowed to make a response, additional evidence was presented until the monkey made a response (or until 1,500 ms had elapsed in which case the trial was aborted). Accordingly, the task is best viewed as a reaction-time task with a minimum viewing duration. We have added this explanation to the legend of Figure 1.

A difference in RT of ~100ms between the easy and difficult conditions is a large effect (in the random-dot motion discrimination task, the average integration time is ~250 ms; Shadlen et al., Science 2016), because it occurred even after 500 ms of stimulus viewing. This finding is not compatible with a deadline (or quickly collapsing bound) and implies that the criterion to terminate decisions depended on the state of accumulated evidence. In support of this interpretation, in the original publication (Figure S1D, Lorteije et al., 2015), we showed that the psychophysical kernels were protracted in trials where the monkeys took longer to respond. In contrast, an external-cue or a time-based strategy predicts that response times should be independent of difficulty and that the kernels should be identical for fast and slow responses. Of course, we cannot claim that a “quantitative match between the model and data is truly impossible”. But if a solution exists it would be more complicated that the one postulated by H and M.

[Editors' note: further revisions were requested prior to acceptance, as described below.]

*There are two points in particular that we think might benefit from further discussion/analyses in your paper. Specifically:*

*1) As you have indicated, the behavioral data show that stimulus difficulty at the L1 branch did not influence performance at the L2 branch. In the original Lorteije et al. paper, a flat model without lateral interactions did not reproduce this result. Hyafil and Moreno Bote then showed that it could be reproduced with a flat model with lateral interactions and rectification. You have pointed out that implementation required very high decision bounds that would have produced unrealistic RTs. Hyafil and Moreno Bote now produce a version of the model that reproduces the L1 dependence and realistic mean RTs, using a kind of "collapsing bound" that is infinite (unreachable) for the first several samples that appears to reasonably account for the task design that allowed responses only after 500 ms of stimulus viewing.*

*2) Both papers discuss the new analysis by Zylberberg et al. showing that the strength of evidence at L1 affects V4 activity elicited by the L2 and L2' branches. Both papers are in agreement with the finding but disagree with the interpretation. Part of the disagreement appears to stem from misunderstandings from some comments in an earlier version of the Hyafil and Moreno Bote manuscript. Specifically, as quoted by Zylberberg et al., Hyafil and Moreno Bote state in Appendix A that the flat model predicts that "selection signals at levels 2 are only influenced by information provided at level 2 branches […] and not by information provided at level 1." The new analyses by Zylberberg et al. clearly contradict this prediction. However, in their main text, Hyafil and Moreno Bote stated that "this observation is most compatible with the flat model and by itself rules out the hierarchical model that relies on complete neural segregation of integration of L1 and L2 evidence." We asked Hyafil and Moreno Bote to reconcile these statements. In the new manuscript, they provide the following comment: "At this point, an important distinction has to be made in the hierarchical model between localized activations, which indeed mix evidence from both branches, and selection signals at L2, extracted by looking at the difference between two units in the same L1 branch, and which as shown above are largely insensitive to L1 signals."*

In our revision, we adjusted our manuscript to address the changes made by Hyafil and Moreno-Bote in their last revision. Just as the previous versions of their ‘flat’ model, their new model with collapsing decision bounds predicts a tradeoff between the accuracy at L1 and L2 that we do not observe in the data. As in our previous submission, we demonstrate that decision making can be remarkably flexible. Subjects can switch from parallel to serial strategies, which is impossible for flat models.

We have now had to address a number of versions of the commentary by H and M, which has become a moving target. We were therefore reassured with your e-mail from Nov 10^th^ where you indicated that this is the final round of revisions. In that e-mail you also indicated that the editors may want to suggest small changes to one or both titles or abstracts for consistency. We would indeed appreciate it if the misleading title of the paper by H and M would change, because their paper did not “break down” hierarchies of decision making. In fact, we believe that these interactions bolstered the case for hierarchical decision making strategies in the primate brain.